# Practices of Self-Care in Healthy Old Age: A Field Study

**DOI:** 10.3390/geriatrics8030054

**Published:** 2023-05-13

**Authors:** Estela González-González, Carmen Requena

**Affiliations:** Department of Psychology, Sociology and Philosophy, University of León, 24004 Leon, Spain

**Keywords:** self-care, older adults, aging, friendly, development

## Abstract

Two competing psychological approaches for how to care for oneself to stay healthy in old age have coexisted and dominated the scientific literature. Objective: Identify the self-care practices of healthy older adults and establish the relationship between these practices and the cognitive processes involved. Method: 105 healthy older people (83.91% women) recorded their self-care practices using the Care Time Test and underwent a cognitive evaluation. Results: The frequency and variety of different activities that participants spent performing on a day of the week where they had the fewest obligations are as follows: nearly 7 h on seven survival activities, 4 h and 30 min on three maintenance of functional independence activities and 1 h on one activity that promoted personal development. Older people who carry out activities in a developmental approach showed better everyday memory (8.63 points) and attention levels (7.00 points) than older people who carry out activities using a conservative approach (memory: 7.43; attention level: 6.40). Conclusion: The results evidenced that the frequency and variety of activities that promote personal development are associated with better attention and memory performance.

## 1. Introduction

Care is an essential element of our existence, reflected in most cultures in colloquial expressions of politeness, such as “take care” (“cuídate” in Spanish, “aufpassen” in Germany or “当心” in Chinese). Given the central position we assign to care in our lives and its prominent role in our survival and personal development, one would expect care to occupy an important space in the health and social science fields. In fact, a theoretical understanding of what it means to care and how to care has often been claimed as an essential element among health professionals but not among social science professionals [1,2]. Indeed, several nursing theories of self-care continue to be taught as part of nursing degree training programs around the world. However, some of them, such as Orem’s [3] Self-Care theory or Watson’s [4] Science of Human Caring, have been criticized for providing little guidance on how care can be achieved in practice [5]. Furthermore, this unidisciplinary perspective has posed some limitations to fully explaining the complex and dynamic factors related to psychosocial contexts while also serving to perpetuate the thinking that care practices are associated with illness [6].

In 21st-century society, two major developments justify the revival of the study of care in the field of social sciences, specifically the scientific field of the psychology of old age. On the one hand, population aging is about to become one of the most significant social transformations of this century. According to the “*World Population Prospects 2022*”, people aged 65 and over accounted for 9.54% of the world’s population [7]. In absolute numbers, 761 million people worldwide were 65 years old last year, a figure that will increase to 1.6 billion in 2050 [7]. This unstoppable trend of aging societies in most countries presents an opportunity to exchange the concept of older people as passive subjects of care with a proactive model of old age that ensures a co-responsibility for care, more in line with the “new old”. On the other hand, the experience of COVID-19 has accentuated the need for a paradigm shift that considers care to be not only a right but also a duty that concerns all citizens. In short, care in the social sciences is defined as the practices that must be deliberately carried out by each individual in order not only to stay healthy but also to continue their own personal development [8].

The concept of self-care is arguably related to healthy aging, as promulgated by life-span psychology. In recent decades, two competing psychological approaches to how to care for oneself successfully in old age have coexisted and dominated the scientific literature [9]. On the one hand, a conservative approach promotes undertaking physical, cognitive and social activities to remain fit in old age. The dominant discourse of this approach is that, to a large extent, old age is a period of loss mitigation; new and uncertain situations are avoided so that one will avoid making mistakes [10,11]. Consistent with this, authors such as Baltes explain how older people use cognitive strategies of selection, optimization and compensation to cope with age-related losses. Baltes’ postulate promotes the popular notion of “activism”, which is characterized by being physically and cognitively active (or fit) in familiar or static environments. Simultaneously, other authors have complemented this approach by introducing the spiritual factor. Therefore, this model comprises four dimensions: physical, cognitive, social, and spiritual [12]. Spirituality/religion is related to the personal meaning of life and its relationship to a supreme being, and although it is a complex construct, spirituality seems to be related to healthy aging behaviors, such as self-care [13]. On the other hand, in recent years, a new approach to healthy aging postulates that personal development in old age is possible [14]. Over time, previous knowledge becomes irrelevant or obsolete; thus, older people must acquire new knowledge to continue functioning autonomously. Adaptation to changes in one’s environment, especially those related to technological developments, has become a necessity as compared to a few decades ago [15]. Therefore, in contrast to the compensatory models of coping in old age, the developmental approach promotes participation in new activities in dynamic environments. The notion of activity in this approach is synonymous with control and “staying sharp” as opposed to “staying active”, which implies performing activities in a static environment [16]. Research on retired people suggests that people who follow the developmental aging model maintain and even improve their cognitive abilities for longer than those who do not [17].

Some researchers highlight that, despite some cultural idiosyncrasies in the perception of self-care, there is a cross-cultural consensus on the characteristics considered most important when referring to someone who is aging healthily. Most ethnic groups mention being in good health, having mobility, independent living, being socially active and participating in cognitive activities. On the other hand, when older people are asked directly about what it means to take good care of oneself, they refer not only to the practice of physical, cognitive and social activities but also to “religious practice”,“continuing to engage with the world and with others” and “not feeling lonely or isolated”. Based on the above, it can be stated that two perspectives of healthy aging coexist: one that is aimed at preserving health to maintain physical, cognitive and social functional independence, and sometimes spirituality, and the other is linked to the practice of new activities in dynamic environments. What remains unclear from previous research is how both conceptualizations of healthy aging relate to self-care practices and cognitive processes in old age. Therefore, this research aims to (a) identify the self-care practices of healthy older adults who practice a healthy lifestyle and (b) establish the relationship between self-care practices and the cognitive processes involved in self-care practices in daily living. In order to do so, the amount of time spent on and the variety of self-care activities will be recorded. This measure has been proposed by Bielak et al. [16] as a suitable metric that predicts cognitive functioning. The starting hypothesis is that the self-care activities that predominate in healthy older people are related to personal development. A second hypothesis is that self-care activities that promote personal development have a significant positive impact on the cognitive processes involved in daily living. A third hypothesis is that both frequency and variety predict the association between self-care practices and cognitive processes in old age.

## 2. Materials and Methods

The use of a cross-sectional observational design, which includes self-care practices in everyday life.

### 2.1. Participants

This study was carried out with a convenience sample recruited from senior centers in the central and peripheral areas of the City of León (Spain). Specifically, 105 healthy older people, who live at home and attend different senior centers to participate in activities that promote functional independence and personal development, were recruited. These senior centers follow the strategic plan of public policies for the self-care of aging, regulated by the Government of Spain [18]. In addition, this strategic planning is inspired by the design of programs and policies from a healthy aging approach recommended by the WHO [19].

Inclusion criteria: (a) people over the age of 60; (b) living at home; (c) independent in activities of daily living. Exclusion criteria: (a) people with neurological or psychiatric diagnoses and (b) people with physical or sensory limitations that prevent them from following instructions.

### 2.2. Assessment Instruments

Rivermead Behavioral Memory Test (RBMT) [20]. This test is used to assess the different memories involved in everyday life. It also assesses functional capacity and the follow-up from daily memory stimulation programs. It is a short, ecologically valid test, about 30 min long, consisting of 12 subtests. In Spain, the original study was translated into Spanish [21] and psychometrically validated [22] with healthy subjects over 70 years of age to categorize the profile scores. The present study used the score profile proposed by Requena et al. [23], adjusted for age and education level.

The Rule Change Test is a subtest of the Behavioral Assessment of the Dysexecutive Syndrome (BADS) battery [24], in which the skills and demands involved in everyday life are assessed. Mental flexibility and the ability to inhibit an automatic response are also assessed. The profile score is obtained as a function of the number of errors and the time spent completing the test. The test has adequate psychometric properties in its Spanish version [25] and has been widely used in research.

Older Americans Resources and Services (OARS) [26]. This instrument provides information on the social resources of older adults. The Spanish version by Grau et al. [27] shows psychometric characteristics equivalent to the original. It is a 10-item instrument that is easy to apply and provides information on social interaction in family and social settings. The duration of the test is approximately 10 min. 

The Time of Self-Care Test. The test depicts a clock face divided into 24 slices, representing the 24 h of the day. In each of the 24-h slots, participants, by interviews, recorded the amount of time they spent on self-care activities. This measure records the amount of time and the variety of activities related to survival (basic activities of daily living (BADL) and instrumental activities of daily living (IADL)), maintenance of functional independence (physical, cognitive, social and spiritual), and personal development (reflection, technological and new activities). These metrics were chosen because Bielak et al. [16] found evidence that a greater variety of activities protects against cognitive impairment. The frequency of a specific activity was calculated as the total time one spent doing a particular activity on the day of the week they had the fewest obligations. Variety was calculated by counting how many different activities of each type were performed for at least 15 min in a single day. In addition, subjects were specifically asked to indicate the types of hobbies they engaged in, any new activities that they had started one year prior, and the amount of time they spent on self-reflection.

### 2.3. Procedure

Our data collection was conducted between September 2021 and March 2022. Participants were recruited by placing information posters on the notice boards of day centers and through information sessions held by the researchers in day centers. The older people volunteers provided a contact telephone number through which the researchers arranged the volunteers’ care assessment appointments. Please note that the study was conducted at the day centers so that the older people would feel like they were in a familiar place. 

All participants were informed of the research conditions, specifically that they were free to leave the study at any time. The subjects signed the study’s informed consent before being assessed. The study followed the Helsinki Declaration, and the Ethics Committee of the University of León approved the protocol (ref. 0225).

### 2.4. Statistical Analysis

Statistical processing was carried out with the SPSS v.28.0 statistical software. In particular, sociodemographic data were collected, and the scores obtained from the applied tests were calculated. The statistical treatment was carried out with the SPSS v.28.0 statistical program. A Pearson correlation analysis (r) was performed between the self-care practices and the cognitive processes involved in the tasks of daily living performed by the older person. The mean differences were calculated using the Student’s *t*-test for the sociodemographic variables (age, sex, schooling, convivence, confidence in others, feelings of loneliness and the number of medical visits) and self-care practices (survival activities, maintenance of functional independence and personal development). The significance level was *p* < 0.05.

## 3. Results

The sample comprised 105 healthy older adults; most (83.81%) were women with a mean age of 74.20 (±6.74) years. Most older adults in this sample had basic studies (55.20%), and the remaining 44.80% had advanced studies (eight or more years of study). The percentage of individuals who lived accompanied was 56.20%. The most common frequency of feeling loneliness was a few times (36.20%), followed by quite often (33.33%) and rarely (21.90%). The following rates of medical visits were found: 16.19% for one time a month or more, 45.71% for 4–6 times a month and 38.09%for once a year. Older adults spend almost 18 h inside the home, of which 8 h and 30 min are for sleeping and nearly 7 are for staying sitting (half of this time are for watching television and using technological devices). The scores in cognitive processing are normality (7.93 (±2.28) for everyday memory and 6.65 (±1.45) for the attentional level). 

Table 1 displays the amount of time and variety average of the self-care practices carried out by healthy older adults. Participants engaged in a higher frequency and variety of survival activities than activities for maintaining functional independence and personal development. Healthy older adults dedicated more than a quarter of the day to carrying out over seven activities of survival; 17.66% of the day was intended for nearly three maintenance activities and only an hour for one personal development activity. The time interval between the activity most practiced (IADL) and least practiced (reflection activity) was 3 h and 15 min. Furthermore, the variety of activities ranged between a maximum of seven (survival) and a minimum of one (personal development).

The Appendix A describes the frequency and variability of the self-care activities according to the classification of the conservative approach (Appendix A) and the developmental approach Appendix A specifies the frequency and variability of BADL and IADL, which comprehend survival activities. Appendix A shows the frequency and variability of the physical, cognitive, social, and spiritual activities focused on the maintenance of functional independence. Appendix A details the reflective, technological and new activities grouped under personal development. There were statistically significant differences in the self-care practices among healthy older people according to the sociodemographic variables. Women performed a higher amount of time and variety of IADL. Likewise, those participants who had someone to confide in carried out BADL with less frequency. Regarding the maintenance activities, the youngest group and men conducted more physical activities than the oldest group and women. Healthy older adults, who had an advanced educational level, carried out cognitive activities more frequently than people who had a basic educational level. Moreover, men and older adults who had no one to confide in spent more time on reflection activities than women and participants who had someone to confide in (see Appendix A).

Table 2 depicts a comparison among the mean scores of the RBMT and BADS for healthy older adults according to the practice or no maintenance of functional independence and personal developmental variables. In a conservative approach, everyday memory and attentional levels were significantly higher in older adults who practiced physical activities and in older adults who did not carry out spiritual activities (*p* < 0.005). There was no difference in the cognitive processing domains for cognitive and social activities. 

Regarding the activities related to the developmental approach, participants who performed both new and technological activities obtained significantly higher scores in everyday memory and attentional levels than older people who did not have these types of practices. Even more, the scores obtained by these participants were associated with an optimal level of everyday memory. There was no difference in the cognitive processing domains for reflection activities among groups who practiced it and groups who did not practice it.

Table 3 shows the correlations between the variety and frequency of self-care activities using a conservative approach vs. a developmental approach and cognitive processes. These data show a significant negative correlation between the frequency of BADL and everyday memory activities (*p* < 0.005). Older people who spend more time on activities, such as feeding, grooming, or dressing, show worse everyday memory performance. Moreover, the frequency of physical activities shows a significant positive correlation with cognitive attentional processing (*p* < 0.05). Furthermore, the frequency of all maintenance activities provides evidence of a significant positive correlation between these activities and everyday memory and attentional levels (*p* < 0.05). Importantly, both frequency (*p* < 0.05) and variety (*p* < 0.005) show positive correlates with daily memory and attention levels within the developmental approach. Thus, the frequency and variety of the activities are valid metrics for linking self-care practices with cognitive performance.

## 4. Discussion

The present research estimated the participation of healthy older adults in self-care practices, categorized into survival, the maintenance of functional independence and the promotion of personal development. Furthermore, it evaluated the relationship between self-care practices in the cognitive processing of everyday memory and the attentional level of healthy older adults living in the community. The results provide evidence that older adults who spend most of the day carrying out a great variety of survival activities had fewer proportions of performing maintenance of functional independence activities and dedicated only an hour to conducting activities promoting personal development. Nevertheless, reflection, technological and new activities, together with physical activities, are associated with better everyday memory and attentional levels in healthy older adults. Moreover, the results provide evidence that the frequency and variety of activities are valid metrics for identifying the relationship between self-care practices that promote personal development and cognitive processing.

Throughout the 20th century, life expectancy increased by about 10 years for each generation, which means that a 75-year-old today has the same mortality rate as a 65-year-old in 1950 [28]. Thus, based on the hypothesis that 75 is the new 65, we did not expect to find differences between the two age groups in the practices of self-care activities [29]. In fact, this research shows that the age group younger than 75 years has habits similar to those older than 75 years, except that the latter practice physical activity less frequently (Table 1). As for the sex characteristics, the number of female participants is three times that of male participants. This trend is common due to the fact that the older adult population consists mainly of women, as they live longer on average than men. In 2022, worldwide, there were 85 men for every 100 women in the 60+ age group [7]. Furthermore, some researchers claim that this gender imbalance in social participation has accumulated throughout life [30]. Typically, during adulthood, women had to take care of the household chores and family care and therefore had little time for personal development activities. At the same time, men in adulthood spend most of their time outside the home, working and socializing with their peers. In old age, however, the distribution of men’s and women’s time is reversed. While women face this stage as an opportunity to get out of the house, expand their social network and, in short, take care of themselves, men face old age with a less socially participative attitude [31]. However, the results of this research show that women continue to dedicate significantly more time to doing household chores than other types of activities, while men devote more time to reflection activities. Both men and women performed more frequently in physical rather than cognitive, social or personal development activities. This preference may be influenced by the dissemination of the benefits of physical exercise in social and health contexts. Physical exercise that is adapted for the older population is called the “anti-aging pill” because it is the most effective non-pharmacological treatment for preserving independent living, par excellence [32]. Physical exercise directly maintains and improves musculoskeletal, osteoarticular, cardio-circulatory, respiratory and psycho-neurological functions. Indirectly, physical exercise has a beneficial effect on functionality, which is synonymous with better health, better adaptive responses and increased resistance to disease [33,34].

Participants with an advanced level of education carry out cognitive, technological and reflective activities and, in turn, show better attention levels and improved memory. These results are consistent with those found by experts who recommend the practice of cognitive stimulation to preserve or even improve mental capacity, especially after retirement [35]. Some researchers link the positive effect of cognitive activities to the existence of cognitive reserve [36]. Social activities are also associated with a lower risk of cognitive decline in older people [37]. The study’s participants who live with others spend more time on social activities and score higher on memory and attention tests than those who are lonely or do not have anyone they can confide in. According to the research reviewed, family relationships and interactions with peers through participation in social activities not only provide emotional and social support but also improve cognitive function [38].

The location where participants carry out the activities informs us about their capacity for self-determination [39]. The participants spent 75% of their time at home, which provides insight into how they manage self-care time at home. In the present study, the participants spent most of their time resting (sleeping) and sitting in front of a screen (TV or electronic device). It is noteworthy that the group of participants who used technology did so on social media platforms, such as WhatsApp or Facebook. These types of resources, offered by the Internet, contribute to the social integration of participants with their peer groups and family members, which is essential for ensuring their social development [40]. Moreover, the Internet offers possibilities for staying up to date with and acquiring new knowledge. In this way, the participants who showed an interest in promoting good self-care practices by using technology obtained better scores in the cognitive function tests.

The frequency metric of time dedicated to BADL (but not variety) negatively correlates with cognitive processing in daily living. Thus, the higher the frequency of BADL, the lesser the cognitive level of the participants. Similar results were encounter from other studies, the frequency and variety of physical activities assist in the maintenance of functional independence; likewise, technology and new activities provide evidence of a positive relationship with everyday memory and attentional level performance. However, most research focusing on the relationship between activity engagement and cognition uses frequency as the main metric [16]. Indeed, the total time spent on a wide range of activities over a day, a week, a month or a year is the most common way to assess a participant’s engagement in a healthy lifestyle [41]. However, there is no clear conclusion in the literature that the frequency of activity is a more sensitive metric than variety in assessing associations with cognitive performance in older age.

It is worth pointing out some limitations and future directions of this research. The first limitation is related to the type of sample used in this study. The participants were healthy, engaged and likely not representative of all older adults. A second limitation is related to the nature of the cross-sectional data, which makes all the relational statements purely tentative and is based on the robustness of the dimensions of the amount of time spent on activities and the different types of leisure activities undertaken. Therefore, similar models should be tested in larger samples and other contexts to compare the results and explore the generality of the results.

## 5. Conclusions

This study finds evidence that a variety of metrics are adequate for assessing the relationship between maintenance activities and cognitive performance. Both frequency and variety are valid metrics for identifying the relationship between self-care that promotes personal development and everyday memory and attentional levels. Furthermore, the results suggest that performing physical, technological and new activities could be the objective of public policies aimed at promoting the self-care of healthy older adults living in the community.

## Figures and Tables

**Table 1 geriatrics-08-00054-t001:** Mean frequency and variety of self-care practices.

		Frequency		Variety	
	N	∑	SD	∑	SD
CONSERVATIVE APPROACH					
Survival activities					
Basic activities of daily living (BADL)	105	2.56	0.48	4.03	0.66
Instrumental activities of daily living (IADL)	101	3.55	1.33	3.20	1.04
Total		6. 44	1.47	7.24	1.24
Maintenance of functional independence activities					
Physical activities	98	2.43	1.26	1.35	0.56
Cognitive activities	63	1.57	1.02	1.83	0.94
Social activities	43	1.22	0.36	1.09	0.29
Spiritual activities	13	1.07	0.43	1.33	0.65
Total		4.24	1.55	2.97	1.63
DEVELOPMENTAL APPROACH					
Personal developmental activities					
Reflection activities	29	0.43	0.32	1.00	0.00
Technological activities	23	1.11	0.48	1.04	0.37
New activities	45	1.34	0.55	1.33	0.64
Total		1.07	1.25	1.08	1.14

Note: BADL = basic activity of daily living (feeding, grooming and dressing); AIDL = instrumental activity of daily living (housekeeping activities, running errands and transportation); physical activities (strolling, gardening, swimming and gym); cognitive activities (reading, listening to the radio, writing and memory training); social activities (going out with family and friends, helping family and friends and eating out with family and friends); spiritual activities (going to the church, praying and religious festivities); reflection activities (meditation, writing in a journal, and review of the day); new activities (volunteering, joining an NGO and planning a trip); technological activities (social networks, web surfing and using a computer).

**Table 2 geriatrics-08-00054-t002:** Mean scores on cognitive processing as a function of the practice of self-care activities.

		N	Everyday Memory			Attentional Level		
			∑	SD	t	∑	SD	t
CONSERVATIVE APPROACH								
Physical activities	Yes	98	8.02	2.14	1.368	6.76	1.43	2.661 **
	No	7	6.87	3.64		5.37	1.19	
Cognitive activities	Yes	63	8.11	2.35	1.346	6.76	1.52	0.923
	No	42	7.56	2.16		6.49	1.34	
Social activities	Yes	43	8.12	2.21	0.684	6.76	1.36	0.598
	No	62	7.81	2.33		6.58	1.52	
Spiritual activities	Yes	12	7.00	1.91	−1.513	5.67	1.30	−2.562 *
	No	37	8.05	2.31		6.78	1.43	
DEVELOPMENTAL APPROACH								
Reflection activities	Yes	29	7.82	2.52	−0.270	6.57	1.53	−0.406
	No	76	7.95	2.22		6.70	1.43	
Technological activities	Yes	23	8.86	1.42	2.197 *	7.36	.95	2.673 **
	No	82	7.68	2.41		6.46	1.51	
New activities	Yes	44	8.63	1.70	2.696 **	7.00	1.15	2.101 *
	No	61	7.43	2.52		6.40	1.60	

Note: everyday memory was assessed with the Rivermead Behavioral Memory Test; the attentional level was assessed with the Behavioral Assessment of the Dysexecutive Syndrome. * *p* < 0.05; ** *p* < 0.005.

**Table 3 geriatrics-08-00054-t003:** Correlation between frequency and variety of self-care practices and cognitive processing.

Activity	Everyday Memory	Attentional Level
CONSERVATIVE APPROACH		
BADL		
Frequency	−0.266 **	−0.102
Variety	−0.156	−0.138
IADL		
Frequency	0.121	0.003
Variety	0.191	0.117
Total survival		
Frequency	0.004	−0.039
Variety	0.081	0.031
Physical activities		
Frequency	0.175	0.225 *
Variety	0.179	0.252 *
Cognitive activities		
Frequency	0.151	0.120
Variety	0.161	0.145
Social activities		
Frequency	0.054	0.051
Variety	0.043	0.055
Spiritual activities		
Frequency	−0.125	−0.189
Variety	−0.080	−0.129
Total maintenance of functional independence activities		
Frequency	0.232 *	0.236 *
Variety	0.184	0.187
DEVELOPMENTAL APPROACH		
Reflection activities		
Frequency	−0.010	−0.024
Variety	−0.011	0.012
Technological activities		
Frequency	0.096	0.226 *
Variety	0.228 *	0.247 *
New activities		
Frequency	0.259 **	0.205 *
Variety	0.145 **	0.134 *
Total developmental activities		
Frequency	0.197 *	0.240 *
Variety	0.260 **	0.239 *

Note: everyday memory was assessed with the Rivermead Behavioral Memory Test; the attentional level was assessed with the Behavioral Assessment of the Dysexecutive Syndrome. * *p* < 0.05; ** *p* < 0.005.

## Data Availability

Research data are publicly available: https://buleria.unileon.es (accessed on 13 March 2023).

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
