# Peer review of "Practices of Self-Care in Healthy Old Age: A Field Study"

_geriatrics, 2023, doi:10.3390/geriatrics8030054_

Round 1

Reviewer 1 Report

Thank you for the opportunity to review the manuscript titled, “Friendly practices of self-care in old-age: a field study.”  Overall I found this study very interesting and a contribution to available knowledge.  I would however recommend some edits to all sections of the paper to improve clarity. My specific comments are below.

Abstract: The objective could be stated more clearly. I’m less clear that this work is really about survival. I appreciated the objective as written in the discussion p.16 294-297. I’m unclear what friendly self-care refers to. Be more specific that the model was a regression.

Introduction: I think that the introduction does a nice job of summarizing the sociology of care, however I believe a more targeted introduction focusing on the topic of self-care and importance of studying it would be of more value to the reader.  Also, it is unclear what this study adds to the current literature/state of knowledge?  It clearly is important work, but it could be better set up in the introduction to support its importance. Also please be consistent with the aims/objectives statements used through the paper they vary a bit in what the actual goal is.

Materials and Methods: Please provide slightly more overview before jumping into the specific sections.  We these in-person interviews done at one time--when?

It was unclear to me how measures were derived from the assessment instruments. Please link them up (what is listed in Table 1) to assessments and then also domains of interest.

In the models could you say more about how survival is being used/defined here?  This was an issue for me throughout the manuscript.  You mention caregiving practices as being what was focused on in the model. Why? It seems more broad in Table 5? This was unclear to me. Please clarify rationale for running the regressions you did.

It was unclear what you may have done with any missing values.

Results: You have a lot of very long tables.  I wonder if the less important ones could be moved to supplemental materials.  I would like to see more write up of the statistically significant variables in comparison to what is there. 

Sometimes there are sig. differences, yet they are quite small. Are these “clinically meaningful”?

Table 1 title could be improved to include all content.

p.6 lines 198-204: I would like more description of the statistically significant findings beyond what is listed.  In particular, perhaps presenting the results where there is a larger threshold between times.  Many times don’t seem that different to me.

p.9 lines 218-222: Be more specific about which measures within the concepts were statistically significant.

p.15 line 254: You mention total activities, what does this include?

Discussion:

P15 lines 264-266: The sentence starting with in particular feels like too far of a jump from what the findings are showing to me.  It is unclear that the participating older adults felt their participation changed stereotypical images of old age. This work does help us understand type of activities they are undertaking and time spent in them which may help further our understanding and assist with ways to develop ways to support older adults self-care efforts in the future.

I would like more discussion on the clinical meaningfulness of the work.

In the limitations section, I would highlight that participants were healthy and engaged and likely not representative of all older adults. I would also add that conducting the assessments in the health centers was convenient but may not have been the setting of choice for the older adult and may have influenced responses. Would you expect any recall challenges about time spent in certain activities which may influence findings? 

Author Response

Please see the attachment,

Reviewer 2 Report

REVIEWER: I recommend a major revision. 

The topic of the manuscript is very interesting since it aims to search for proactive actions of care for the elderly which could therefore promote active and healthy aging. Nevertheless, there are some questions (I will enumerate them in order below) that should be adequately solved before this manuscript could be published. 

English spelling and grammar should be revised throughout the text.

Introduction: page 1, paragraphs 37-38. The statement “ageing necessarily implies dependency” is not correct. Although the percentage of dependent people is higher as the population gets older, ageing itself does not imply necessarily dependency. 

2.1. Participants: please, indicate the procedure used to determine the adequateness of the sample. 

2.2. Assessment instruments: in the tables are cited instruments of daily living activities and advanced activities that are not explained in the section. 

Revise in all the instruments their psychometric information since it is only mentioned in the NHP instrument. 

2.4. Statistical analysis. Page 4 paragraphs 186-187. Revise and correct the following: was at 5% (p < .05) and 5% (p < .005).

Results

In my opinion, information regarding the socio-demographic variables of the sample should go to the participant's section.

About the results regarding table 1: the information given in the results section is too short to understand the table. Please, explain adequately the statement “However, participants’ health level negatively affects these data”. In which sense? to what extent? What implications has this fact had on the results?

Table 1: Variables memory, executive function, and health: please, explain to what instruments they relate.  

The authors divide participants regarding age into two groups (<75 and >75). Please, give an explanation for this specific split. In education level, the authors mentioned basic and advanced but no explanation regarding what this implies is given. 

The table shows significant differences between variables but with the information given, I do not understand what it refers to. In the results section there is no explanation of the data. Also, it is not mentioned whether a T student or an ANOVA has been conducted (F value or T value (depending on the analysis carried out) should be included). With this lack of information in the text and in the table I cannot understand the results shown. Besides, and as another example of this flaw, a p significance between the age groups is indicated in both cases. In my opinion, the difference is between the groups so I do not understand why the * value is given in all the categories. In the variables where more than two groups are compared, the differences between all the groups should be done by post-hoc analyses. No information is given in relation to this in the text or in the table so the reader cannot know between what groups the differences are. In the foot of the table, there is a mistake (p should be < than .05, otherwise it wouldn´t be significant). 

The flaws I stated above about table 1 (table and the explanation of the results in the text) are the same in table 2, table 3, table 4, and table 5. 

Also, in table 2 and table 4, the authors should indicate what the variables Physical, cognitive, social, technological, spiritual, and reflection exactly mean. 

On page 15, paragraph 243 the authors mention supplementary material for tables 1 and 2. This information should be placed where tables 1 and 2 are explained. 

Discussion and Conclusions:

In my opinion, the discussion and conclusions given by the authors could only be reached once the results section is properly redone. 

References: the authors do not follow the journal guidelines. For example, the words vol. and pp. are included when they should not be cited; the volume should be in italics. 

Reviewer 3 Report

The introduction includes some intriguing background on ‘caring’ but does not quite set up the paper effectively. I was looking for a definition of ‘self-care’ that would framework the research questions. Without this framework the reader has to construct a connection between ‘self care’ and the items that are measured.  Please specify what you define as a self-care practice. Also, please define or replace the term ‘friendly activity’ as it isn’t a term I am familiar with in the literature. 

There is a statement about grouping the respondents: 

“Participants were mainly characterized into the following groups: not having to take care of others, not taking medication, not engaging in reflective activities, and having internet at home.” Why were these grouping made? What makes this useful? The group does not appear elsewhere in the paper, I was expecting to see the tables organized based on this.  

Table 1 has a few issues: – looks like the data for >75 and <75 are flipped. What is the difference between ‘convivance’ and alone versus accompanied and also ‘people in the home’? These variables are cross tabbed, but seems like they measure the same thing so it is hard to interpret. 

The tables 3, 4 and 5 are very data dense and it is difficult for the reader to exact meaningful information. Please spend more time in the text processing the data for the reader and explaining what correlations appear. Consider moving the tables to a supplement and providing higher level summaries of the data tables.

It is clear that the team performed many statistical analyses, but the text does not provide sufficient explanation of the hypotheses that drove the analysis or definition of the variables. It appears the some of the variables were constructed from multiple data points, but that isn’t clear. For example, what activities were counted as BADL? What is the difference between a spiritual and a reflection activity? There is reference to a ‘self-care model’ but there is no explanation of the model.  This model is key to the paper and yet is never addressed. 

Despite reading through the paper carefully two times I am unable pull any recommendations or conclusions out of the research. It was not clear what the outcome of interest is. “Overall, our results are consistent with the understanding of self-care as a healthy way of life that involves being physically and mentally active as well as maintaining social contacts. In particular, the measures of time and variability of friendly-activities had relevant and expected effects on self-care practices.”  I presume I am missing some detail here, because this reads as being very circular to me. As best I can tell you have defined ‘friendly activities’ as being self-care and then report to have found that self-care is predicted by ‘friendly activities.’ It needs to be very clear what variables/activities are effecting what outcome/end state. 

While the paper acknowledges the limitation of the sample, I think there should be discussion about how this sample includes only people who already participate in activities at the senior center, so it represents a group who are already active (seemingly a form of good self-care)  – this is potentially very different from the seniors who do not go to the senior center.  

You have a rich set of data, but need to sharpen the narrative of the paper and more explicit analysis of the results. Tell us the research question you were trying to answer and then tell us what you learned from the survey. Can you tell the reader what activities make for better self-care and why?

Round 2

Reviewer 1 Report

The authors are responsive to my initial comments and the paper has made progress, but still needs additional finetuning before it should be accepted for publication.

The introduction is much improved!

Methods- Setting

I would have liked slightly more information about the senior center where the study was conducted. I only see a participants paragraph.

Results

This section is dense and has too much content.  I think selecting only the most salient results to present is necessary.  There are too many tables and not enough succinct interpretation to figure out the high level learnings most aligned with the objective.

You mention the supplemental materials but don’t give a brief summary of what you found.

Discussion

The discussion is long. What are the main findings and how are they supported in the literature? What is the take away message here?

General

There are spelling errors as well as some sentences which need revision for clarity.  There is a lot of repetition in the writing of the results section when introducing each table.  

I would remove the word elderly and replace it with older persons or older adults.

Reviewer 2 Report

SECOND REVIEW.

Manuscript: practices of self-care in healthy old age: a field study.

The revised version improves the previous one. Nevertheless, there are still some issues that should be modified before its consideration for publication.

Participants:

In my opinion, the information added “convenience sample” does not explain why the size of the sample is sufficient to obtain meaningful conclusions.

Statistical analysis:

In this section, it is said that ANOVA test and Bonferroni post hoc analyses were conducted. The tables show correlations and T-student analyses but no ANOVA  and Bonferroni post hoc analyses are shown.

Results:

Page 4 (194-197) I think that if the authors refer to the information given in a supplementary file, they should include in the text the statistical information needed to support their statement.

Table 1. In my opinion, this table is still very confusing. Some issues:

-         The terms used in the text (static vs Dynamic contexts) do not correspond with the terms conservative and developmental approach (maybe I misunderstood the information)

-          The activities cited in the table are more than the ones in the text.

-         In paragraph 185 the authors say “The data show a significant negative…” The table shows a correlation that is not significant so the conclusion is not valid.

-         In paragraph 187, the authors say “The same non-statistical trend...” On one hand, in the text it is said (although the table does not) that the previous results were significant (they are not) and, on the other hand, if it is not significant, there is no association.

-         In my opinion, it is difficult to follow when the authors are referring in the text to the part “conservative approach” or the data related to the “developmental approach”. I think it would be easier to follow the explanation if they follow the order given in the table and do not mix information. 

-         (216) The authors say “the frequency….is related to self-care” I do not see this data in the table.

-         In the table, once, the terms frecuencia and variablilidad appear (please translate)

-         (216-217) I do not understand if the survival activities cited here are “total survival” cited in the table and before in the explanation. I don´t quite follow what they mean with the part of the sentence: “… or a variety of these measures”.

-         In the note, there is no explanation of the term “total survival” or “total developmental activities”

Page 6(222) If the results are not significant, do not jump to conclusions. Also, the explanation given for tables 2, 3, and 4 refers mostly to the number of activities or times. Although the information is relevant taking into account the topic of the manuscript, it should also inform about the statistical information given in the tables related to the analysis undertaken.

One of my previous question was:

“Also, it is not mentioned whether a T student or an ANOVA has been conducted (F value or T value (depending on the analysis carried out) should be included). With this lack of information in the text and in the table I cannot understand the results shown”. The authors respond “T-value (t student) and F value (ANOVA) has been included in the new tables”.

In my opinion, this respond is not valid since you do not mix both types of analyses. As it can be seen in the tables, T analyses are the only one included.

Table 2. Although in the response the authors clarify the difference between basic and advanced education, it is not present in the table (revise the clarification in the response given since there is a mistake with the symbol < and >).

Page 10 (271-272) Explain here and as a footnote in Table 5 how everyday memory and attention have been assessed.

Discussion

Please, revise the discussion once the results section has been corrected following the previous recommendation

Reviewer 3 Report

The paper is much improved but still has too much extraneous information to make it clear and compelling. In the abstract the conclusion is stated as "the frequency and variability of activities are valid metrics for identifying the relationship between self care practices and cognitive health." This is a very useful and important finding, but it is buried in the paper underneath lengthy descriptions of the multiple linear regression that sought to "determine if the variability of and amount of time spent on survival, personal development and proactive activities could be used as predictors of caregiving practices."   I question the appropriateness of using multiple linear regression and the meaningfulness of the results. I could not find a definition in the paper for 'caregiving practices', but as best I can tell it is an amalgamation of the very activities placed in the regression model. This would explain why the R2 value is so close to 1. I suggest that the authors focus the paper solely on the relationship between activities and cognitive health. 

If the regression results are to remain the paper, then the authors need to provide an explanation of 'caregiving practices' and how this is an appropriate dependent variable, and why there is value in predicting it with the activity information. 

The introduction is much improved and sets up the paper nicely. The Results are quite difficult to interpret and the text from lines 218 to 248 are not helpful- I recommended excluding the regression entirely.  

Check the spelling throughout - see column headers in Table 4 (Technology and Frequency) and in Table 5 'using'. 

Please provide explanation of what you mean by 'having a confidant' and 'having someone to rely on'. I presume these are the same thing, but please define and explain how this was determined and why it matters for your outcomes. 

I don't understand the sentence on lines 285 and 286 about older people sitting down for only 6.4 minutes. Is this for an entire day? How can that be true?

Round 3

Reviewer 1 Report

The authors have addressed my outstanding concerns. Thank you for the opportunity to review the paper!

Reviewer 2 Report

Dear authors,

I think this third version is substantially better than the ones before. The logic of the research is clear from the beginning and the results and discussion are well argued. Congratulations for the improvement. I think the manuscript is now ready to be published.

The reviewer

Reviewer 3 Report

The paper is in great shape. I think this is a good contribution to the literature.